# How Does Religious Faith Impact Positive Youth Outcomes

**Michael A. Goodman * and W. Justin Dyer**

Department of Church History and Doctrine, Brigham Young University, Provo, UT 84602, USA; justindyer@byu.edu
* Correspondence: mike_goodman@byu.edu

**Abstract:** This study investigates the protective aspects of religiosity in young adolescents and explores the potential processes involved. Specifically, we examine the influence of three measures of religiosity—religious salience, intrinsic religiosity, and daily religious experiences—on delinquency and anxiety. The study sample consists of 636 families located in the state of Utah. Additionally, we explore how positive youth development (PYD) constructs may mediate the relationship between religiosity and adolescent outcomes. The findings indicate that religious salience and daily religious experiences are directly and indirectly associated with lower levels of delinquency and anxiety. Furthermore, two PYD constructs—character and connectivity—serve as mediators between religious salience, daily religious experiences, and these outcomes. These findings are discussed within the framework of Bronfenbrenner's PPCT model of bioecological theory.

**Keywords:** adolescent religiosity; positive youth development; spirituality



## 1. Introduction

Many studies have explored the relationship between religiosity or the constructs of positive youth development and adolescent well-being in such areas as academic achievement, self-worth, and internalizing and externalizing problems (see Chadwick and Top 1993; Chiswick and Mirtcheva 2013; Goeke-Morey et al. 2014; Merrill et al. 2005; Pearce and Haynie 2004; Sinha et al. 2007). However fewer studies look at the processes by which religion or the constructs of positive youth development are related to adolescent well-being and fewer yet examine the relationship between religiosity and positive youth development constructs. The purpose of this study is to explore the relationship between these two important developmental constructs and how they impact positive youth outcomes for young adolescents

## 2. Literature Review

### 2.1. Religiosity among Adolescents

Recent studies have provided valuable insights into the prevalence of adolescent religiosity, such as the National Study of Youth and Religion conducted by Christian Smith. According to Smith's report, over 90 percent of American adolescents express a belief in God or some other "cosmic force", while over 80 percent consider religious faith at least somewhat important in their daily and long-term decision-making. Additionally, over 65 percent of American adolescents engage weekly prayer, with a majority praying daily. Regular church attendance is reported by almost 60 percent of adolescents, and the number increases to nearly 70 percent when the decision to attend is left to the adolescent (Smith 2005). Despite evidence indicating a decline in adolescent religiosity (Hardy and Longo 2021), it is clear that religiosity continues to be part of life for most American adolescents. Research has shown that adolescent religiosity positively predicts prosocial behaviors and negatively predicts antisocial behaviors (Stolz et al. 2013). These findings underscore the importance of increasing our understanding of the development of adolescent religiosity.

### 2.2. Adolescent Religiosity and Externalizing Behaviors

Adolescent religiosity tends to correlate with lower externalizing behaviors such as sexual activity, drug and alcohol use, and other delinquent activities. (Edwards et al. 2011; Haglund and Fehring 2010; Landor et al. 2011). The process or reason for the inverse relationship is often not determined though some studies have examined the reasons why. For example, religiosity did not lower an adolescents' desire for sexual pleasure but rather seemed to lead to internalized norms which encouraged youths to delay sexual behavior and place it within committed relationships (Vasilenko et al. 2013). Additionally, religiosity also seemed to foster self-control and both constructs (religiosity and self-control) independently and additively discourage adolescent involvement in risky sexual behavior. (Vazsonyi and Jenkins 2010).

Drug and alcohol use also significantly decreases as adolescent religiosity increases (Bahr et al. 1998; Corwyn and Benda 2000). One study found that religious salience (how important religion is in the life of the subject) mattered less than religious attendance in diminishing drug and alcohol use (Mason and Windle 2002) while another study found the opposite (Corwyn and Benda 2000). Family religiosity, especially family religious attendance has also been found to be protective when it comes to adolescent drug and alcohol use (Foshee and Hollinger 1996; Hardesty and Kirby 1995; Merrill et al. 2005). It was not clear if the reason for these findings indicated that the adolescents internalized their family's religiosity or whether family religiosity influenced parental expectations and resulted in stricter expectation for adolescent compliance.

Finally, family and adolescent religiosity has also been found to correlate with lower levels of juvenile delinquency in school and the community (Li 2014; Pearce and Haynie 2004). Though the reasons for this are not always clear, we know parental religiosity does play a role. Curiously, parental religiosity seemed to be more protective for female adolescence than for males (Regnerus 2003; Regnerus and Burdette 2006). Interestingly, disharmony between a mother and father's religious beliefs, even if both are highly religious, increased the likelihood of adolescent delinquency when compared to parents with homogenous beliefs (Pearce and Haynie 2004).

Though there is a clear correlation between adolescent religiosity and a general reduction in externalizing behavior, when processes or reasons are explored the findings at times conflict or at minimum show a complicated relationship. For example one study indicated that the protective correlation was stronger when considering how an adolescent viewed their relationship with God (relational religiosity) than it was if only looking at adolescents religious activity (Pickering and Vazsonyi 2010). So, an adolescent who felt close to God was less likely to engage in harmful externalizing behavior than one who simply went to church or engaged in other religious activities. However, other studies pointed to the power of those exact activities as the primary reason for the relationship (Moreau et al. 2013). Another study found that the importance of religion to adolescents had little impact on adolescents but church attendance itself did (Mason and Windle 2002). Clearly there is a need for more research to understand the processes and mechanisms involved. Typically, it has been assumed that religiosity impacts externalizing behavior. But the inverse needs to be considered as well, namely how externalizing behavior impacts the development of religiosity. The relationship is likely reciprocal.

### 2.3. Adolescent Religiosity and Internalizing Issues

Adolescents with higher levels of personal and familial religiosity have been found to have increased self-esteem and healthier psychological functioning (Ball et al. 2003). Other studies report that religious teens have better physical and emotional health as well as well-being in comparison with less religiously oriented teens (Chiswick and Mirtcheva 2013). Some research has pointed to shared religiosity leading to lower levels of parental conflict and fewer internalizing problems for adolescents (Brody et al. 1996; Petts 2014; Varon and Riley 1999). Researchers have found that one-way higher religiosity may protect adolescents

from more internalizing problems is through discouraging the kind of externalizing issues listed above which can lead to greater internalizing problems (Ahmed et al. 2011).

Anxiety, depression, and suicidality are faced by many adolescents. Extant research has suggested that adolescents who feel close to God experience less depressive and anxious symptoms (Goeke-Morey et al. 2014; Wright et al. 1993). Another study that sought to understand how religiosity influences internalizing problems found adolescents' psychological well-being to be strongly related with their social experience in religious environments, even more than their church attendance (Pearce et al. 2003). A different study showed that religious attendance rather than socialization had the greater role in better psychological functioning (Ball et al. 2003).

As with externalizing issues, the correlation between religion and less internalizing problems is quite robust, but the details and processes are still not completely clear. For example, in one study greater religiosity was associated with higher self-esteem and better psychological functioning (Ball et al. 2003). This same finding was true of those with higher levels church attendance. However, when looking at religious salience, self-esteem was impacted but psychological distress was not, an interesting apparent contradiction. Another study found that "spirituality", defined as an adolescent's feeling of closeness to God and connection with the transcendent, was not as impactful as more concrete measures such as church attendance and being part of a religious community in relation to internalization issues. Though rarer, some studies approach it from the opposite direction, namely how the internalizing of problems relates to religiosity. The study found that higher levels of internalizing problems predicted a weaker perceived relationship with God a year later (Goeke-Morey et al. 2014). As with externalization and religiosity, more research is necessary to better understand both directionality as well as the specific processes behind the relationships.

### 2.4. Differential Impact of Religiosity on Adolescents

Although as the above literature review indicates, research suggests adolescent religiosity significantly influences several outcomes (usually in a positive direction), this influence likely varies person to person as well as circumstance to circumstance. Bronfenbrenner's (1999) bioecological model proposes that "proximal processes" (i.e., direct, reciprocal, enduring, increasingly complex interactions) influence both developmental and other outcomes. However, "the form, power, content, and direction of the proximal processes affecting development vary systematically as a joint function of the characteristics of the developing person ... [and] the environment" (Bronfenbrenner 1999). In other words, proximal processes will vary in their effects depending on personal and environmental factors. The bioecological model would predict variation in the extent to which adolescent religiosity impacts adolescent outcomes dependent upon (a) presence of proximal processes and (b) the characteristics of the child, and their environment.

Proximal processes represent the initial component within Bronfenbrenner's process-person-context-time (PPCT) model, which provides a framework for comprehending human development. Instead of viewing individual variables as acting independently or solely additively on adolescents, the PPCT model underscores the significance of studying each developmental aspect (proximal process, personal characteristics, external context, and time) concurrently to explore their interactions. Accordingly, it is important to understand how several process-person-context-time constructs may impact adolescent outcomes.

### 3. Current Study

Using Bronfenbrenner's PPCT model, we investigated whether there were differential effects of a proximal processes (daily religious experiences) and several personal characteristics, religious salience and six PYD constructs in two separate spheres of adolescent outcomes: internalizing problems (anxiety for this study) and externalizing problems (delinquency for this study). Based on the PPCT model and the above literature review, we

hypothesized the impact of daily religious experiences will generally be positive but will vary based on several personal characteristics.

## 4. Method

### 4.1. Measures

**Externalizing behaviors.** Nine delinquency-related items adapted from the Child Behavior Checklist Youth Self-Report were used to assess externalizing behaviors (Achenbach 1991; Barber et al. 2005). Both parents and children reported on behaviors such as lying, cheating, stealing, and using alcohol or drugs. Responses ranged from 0 (not true) to 2 (often true), with higher scores representing higher levels of delinquent behavior. Planned missingness was used for this variable with each participant taking 6 of the 9 items. To remove measurement error and handle missing data, a single model with latent variables for parent and child report was created. Thus, information that each provided on externalizing behaviors was included in the model to handle missingness (i.e., parent reports were used to help handle missingness of child reports and vice-a-versa). In factor analyse, five of the nine items were dropped due to poor loadings due to low variance. For instance, the item regarding smoking cigarettes was asked of every child, yet was not endorsed by any (i.e., had no variance). Similarly, only two respondents endorsed that they used alcohol. Factor scores of this model were saved and used in the analysis. The final scale included: disobeying at school, hanging around friends that get in trouble, lying or cheating, and swearing or using dirty language. The measurement model had excellent fit ($\chi^2$(df) = 15.371(18), CFI = 1.00, RMSEA = 0.00).

**Anxiety.** Child anxiety was assessed with the six-item generalized anxiety disorder subscale from the Spence Child Anxiety Inventory (Spence 1998). Participants responded using scale ranging from 0 (never) to 3 (always) with higher scores reflecting greater levels of anxiety. This was modeled as a latent variable within the overall structural model. The measurement model for anxiety fit the data well (CFI = 0.978, RMSEA = 0.054) with all standardized loadings above 0.55.

**Positive Youth Development.** The Positive Youth Development Inventory (PYDI; Arnold et al. 2012) is a collection of 55 Likert scale items that measure changes in levels of positive youth development. This version follows the 5 C's model of youth development (Lerner et al. 2005), by measuring (1) Confidence; (2) Competence; (3) Character; (4) Caring; and (5) Connection. Measurement of a 6th C (Contribution) is included in this instrument. Each item is rated on four-point scale from 1 = Strongly disagree to 4 = Strongly agree. The scale is recommended for ages 12–18.

A measurement model was created for these six domains and factors scores were exported for use in final analyses. The model fit the data well (CFI = 0.938, RMSEA = 0.043). All loadings were above 0.40 with the exception of one item in the confidence factor ("I can do things that make a difference) which loaded at 0.272. However, given the good model fit and in order to maintain consistency with prior work, this item was retained.

**Intrinsic Spirituality.** To measure the degree to which children have an internalized sense of spirituality, Hodge's (2003) six item intrinsic spirituality scale was used. Questions were asked regarding the degree to which children felt spirituality impacts their lives and was important to them. Responses were on a 0–10 scale with 0 indicating that spirituality was less intrinsic and 10 indicating it was more intrinsic. For example, one item asked "Spirituality is:" with 0 being "not part of my life" and 10 being "the master motive of my life, directing every other aspect of my life". A measurement model was created for intrinsic spirituality with the factor score exported for use in structural models. The measurement model fit the data well (CFI = 0.999, RMSEA = 0.030) with all loadings above 0.67.

**Strength of Religious Faith.** The Santa Clara strength of religious faith questionnaire (Plante and Boccaccini 1997) was used as a general measure of religiosity and religious salience. This 10-item scale includes items such as: "I look to my faith as a source of comfort" and "I consider myself active in my faith or church". The measurement model

fit the data well (CFI = 0.997, RMSEA = 0.047; all standardized loadings above 0.82) and exported factor scores were used in final structural models.

**Daily religious Experiences.** A subscale of the NIA/Fetzer Religion and Spirituality scale (Idler et al. 2003) was used to examine the degree to which children felt some connection with God/spirituality on a daily basis with response categories being 1 = "never or almost never" to 6 = "many times a day". Items include "I feel God's presence" and "I am spiritually touched by the beauty of creation". Again, factor scores were exported from a measurement model which fit the data well (CFI = 1.00, RMSEA = 0.00; all standardized loadings above 0.84).

**Controls.** Structural models controlled for race (1 = White, 0 = non-White; unfortunately there was not a sufficient sample size for independent analyses of non-White groups), income, and whether the child was LDS—the primary religious denomination in the sample (1 = LDS, 0 = not LDS).

*4.2. Sample*

Participants were taken from Wave 1 of the Family Foundations of Youth Development project, an ongoing study of adolescents and their families. One emphasis of this study is on faith development for youth from varied religious traditions. However, the first phase begins with a study of adolescents in the Church of Jesus Christ of Latter-day Saints (LDS) with comparison groups coming from those of other faiths and no religious affiliation. Data were collected from two, primarily urban counties in Utah. Families were recruited using the InfoUSA national database which contains over 80 million households across the U.S. Families. Data were obtained on households with children between the ages of 12 and 14. Potential participants were sent letters and follow-up phone calls. One child and one parent filled out the survey with the child being compensated $20 and the parent compensated $30. Of families eligible to participate (had a child between 12 and 14) over 60% agreed to participate. In all, 579 parent-child dyads participated with an additional 40 parents and 17 children who participated with the other not participating (i.e., either the parent or child did not complete most of the survey or did not complete the survey at all) for a total of 636 families represented. These incomplete dyads are included using fill information maximum likelihood (FIML) to handle the missing data. The majority of children were from homes with two married biological parents (92%). Although most children identified as LDS, 14% (n = 80) identified as another religion or no religion. Regarding race, 88% of the sample is White and the other 12% split between Hispanic, "mixed," Asian, Black, and "other".

**5. Analysis Plan**

All analyses were conducted in Mplus 8.0. The model outlined in Figure 1 was fit for anxiety and externalizing behaviors with the Mplus command "Model Indirect" in order to calculate the indirect effects of religiosity/spiritualty on child outcomes. To appropriately calculate standard errors for indirect effects, bootstrapping with 5000 draws were used for externalizing behaviors and anxiety. However, given a limitation in Mplus, bootstrapping cannot be used for the zero-inflated Poisson model and therefore non-bootstrap standard errors are reported. To examine whether gender is a moderator of the relationships we further fit these analyses as multiple group models by gender.

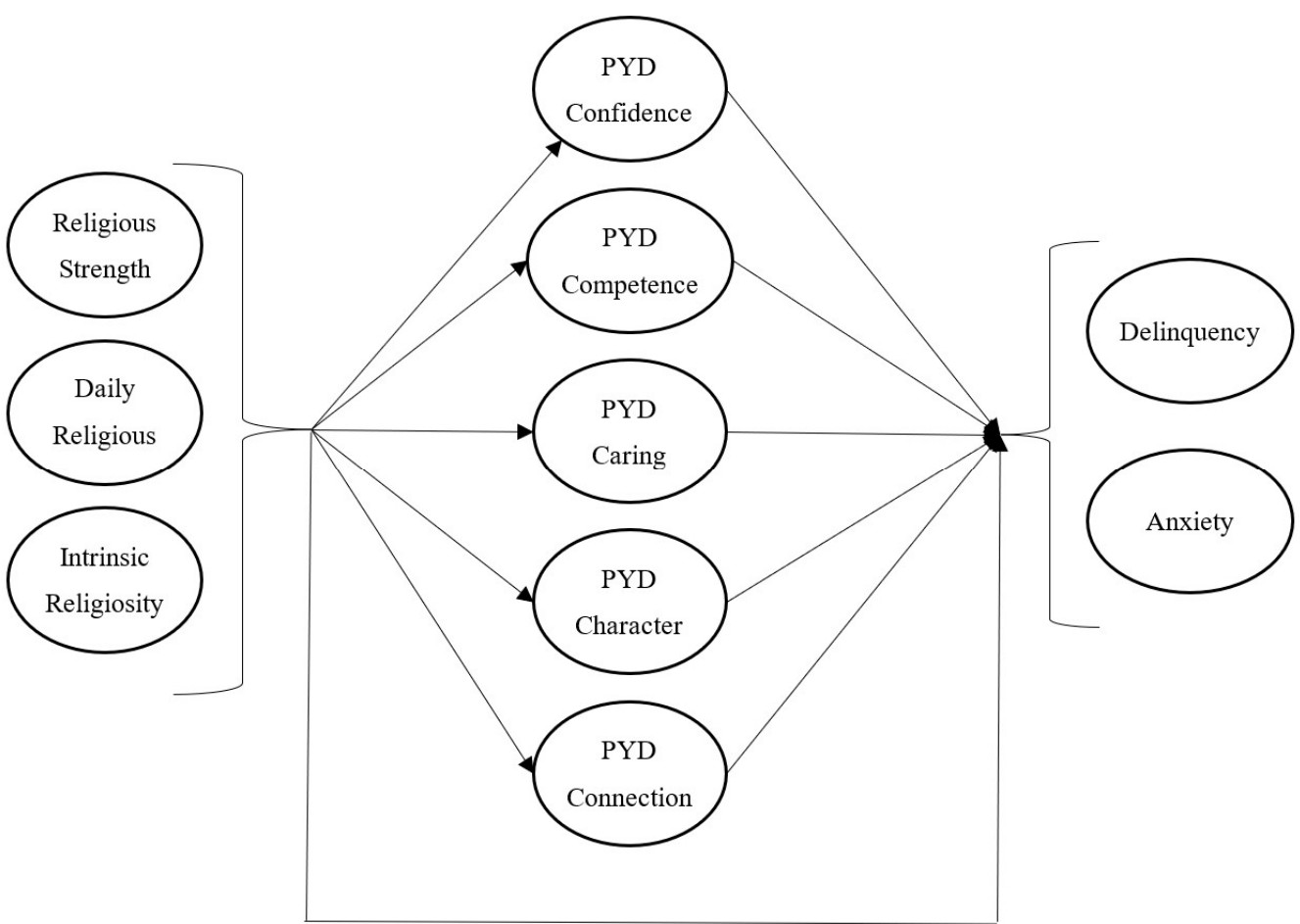

**Figure 1.** Structural Equation Model for Delinquency and Anxiety.

## 6. Results

**Correlations.** Correlations between parent and child reports of video game addiction symptoms and externalizing behaviors were significantly and negatively associated with all measures of positive youth development and religiosity. Anxiety was significantly and negatively correlated with competence, connection, and confidence. It was not correlated with any religiosity variables.

**Anxiety.** For both males and females, competence had a significant, direct effect on anxiety (males: b(se) = −0.21(0.08), *p* < 0.01; females: b(se) = −0.34(0.09), *p* < 0.001). For males, daily religious experiences were related to all six aspects of positive youth development whereas for females daily religious experiences were only related to competence, connection, and confidence. Intrinsic religiosity was only significantly related to confidence for females. For females, religious strength was significantly and positively related to all positive youth development constructs whereas it was only related to competence, caring, and contribution for males (other analysis follow the highly similar pattern of significant and non-significant results for religious/spiritual variables predicting positive youth development). In indirect analyses daily religious experiences was found indirectly related to anxiety through competence for females (b(se) = −0.07(0.04), *p* < 0.05).

**Externalizing behaviors.** No variables significantly predicted either child or parent report of externalizing behaviors for females. For males, connection did significantly predict parent reports of externalizing (b(se) = −0.06(0.03), *p* < 0.05). Yet even though daily religious experiences predicts connection, the indirect effect of daily religious experiences through connection was not significant. However, a single group analysis (both males and females combined) found a significant relationship between connection and externalizing

behaviors and a significant relationship between connection and daily religious experiences. This indirect effect was then significant (b(se) = −0.01(0.00), *p* < 0.05).

## 7. Discussion

### 7.1. Internalizing (Anxiety)

According to a study cited on the website of the National Institute of Health, 25% of adolescents between the ages of 13 and 16 will experience some form of anxiety disorder (National Health Statistics Report 2017). A study cited on the website for the Center for Disease Control estimates the rate at 32% for the same age group (National Health Statistics Report 2017). Clearly many adolescents are impacted by anxiety. This study found that an adolescent's perception of their own competence had a significant direct effect on anxiety for both males and females, where greater feelings of competence lead to lower levels of anxiety. This fits well with Bronfenbrenner's theory since person-level characteristics, such as an individual's perception of their own competence, would be expected to impact how a person would respond to the stressors that might bring on anxiety (Bronfenbrenner 2005b). Daily religious experiences were also negatively related to anxiety in single group analysis but not when the sample was divided by gender (a person-level characteristic) likely due to lack of statistical power. However, daily religious experiences were indirectly associated with lower anxiety by increasing a sense of competence for females but not males. Daily religious experiences fit well within the definition of a proximal process of a meaningful daily experience or interaction. Bronfenbrenner stated that "to be effective, the interaction must occur on a fairly regular basis over extended periods of time" (Bronfenbrenner 2005a).

It makes intuitive sense that adolescents who feel personally competent would tend to experience lower levels of anxiety. However, it would also make sense that youth with strong levels of connectivity (support systems) and confidence would experience less anxiety, but at least statistically, that was not the case for male or female adolescents in this study. Of the three religious constructs tested, only daily religious experiences had a direct significant relationship with lower anxiety in the expected direction. Daily religious experiences were largely defined by regular, daily feelings of connectivity to God. It may be that adolescents who feel close to God find strength and support to deal with stress that other adolescents do not. Religious salience had no significant direct effect but both daily religious experiences and religious salience had a significant indirect effect in single group analysis which was washed out when the sample was divided by gender.

Though religious salience had no direct significant relationship with anxiety, higher levels of it, as well as daily spiritual experiences, both had a significant indirect effect on lower levels of anxiety through the PYD construct of competence when subjects were not grouped by gender. This would seem to indicate a potential channel by which religion can help adolescents cope with the stress by helping them feel more competent. Though all three religious constructs were related to many if not most of the constructs of positive youth development where higher levels of each religious construct were related to higher levels of PYD constructs, they ended up not being significantly related to lower anxiety since the other PYD constructs besides competence ended up insignificant in the model.

### 7.2. Externalizing (Delinquency)

We obtained both parents and child reports of externalizing, delinquent behavior. Using single group analysis (both males and females combined) no relationship was found between any of the religious or PYD variables and child reported delinquency. However, when using parent report of their child's delinquency, a significant relationship was found between connection and delinquent behavior as well as an indirect effect on delinquency from daily religious experiences mediated by the PYD construct of connection. The connection construct measures the strength and importance of an adolescent's relationship with peers, parents, teachers, and others. Adolescents who felt more connectivity were less likely to participate in delinquent behaviors. Logically, it would seem possible for a youth to feel connected to another youth who encouraged delinquent behavior but for this

sample either that wasn't the case or the youth's relationships with parents, teachers and others mediated negative influences by more externalizing peers. Of the three religious constructs, only daily religious experiences impacted externalizing behaviors and that was through connection.

This makes sense since daily religious experiences measure an adolescent's connection to God. Also, one of the questions which makes up the daily religious experiences construct asks if the youth feels, "God's love for me, directly or through others." If the adolescent's connection to parents and others includes people who are spiritually nurturing, the adolescent may feel close to God in part since their parent or other significant individual helps create that feeling. This would fit with Bronfenbrenner's theory of the importance of loving parent-child relationships in positively influencing children's development: "The establishment of a strong mutual emotional attachment leads to internalization of the parent's activities and expressed feelings of affection" (Bronfenbrenner 2005b).

When the sample was divided by gender, the findings became more nuanced. None of the religion or PYD variables predicted child or parent report of delinquent behavior by females. However, this does not appear to be since there was no female delinquent behavior (there was) but since the predictors in our model simply didn't capture the reason behind it. For males, connection did significantly predict parental, but not child, report of externalization. One thing this shows is that parents view delinquency different than the adolescents themselves. But where daily religious experiences did have a significant indirect effort by way of connection for the single group analysis (male and female combined), it did not hold for males. It is possible that daily religious experiences simply do not create the kind of connectivity for boys that they do for girls. This again would make sense within Brofenbrenners PPCT theory that outcomes and development would vary based on both the proximal processes and the person-level characteristics of an individual.

## 8. Limitations and Conclusions

One clear limitation of this study is the lack of sample diversity that would be needed to generalize to other populations. A total of 86% of the sample were Mormon, all were from the state of Utah, and 92% came from intact homes. Mormon adolescents tend to have higher levels of religiosity as measured by church attendance, feeling close to God and lower levels of doubt and disaffiliation (See Smith 2005). This may make the findings most generalizable to other faith traditions such as black and conservative protestant denominations which also tend to have higher levels of religiosity among their adolescent members. Findings will likely vary for single parent families, families from a more diverse geographic population, or families with same sex parents. Also, though there were economical differences within the sample, there was very little poverty. Furthermore, this study looked at younger adolescents from age 12–14. Older adolescents may differ in the processes and extent to which religiosity impacts these outcomes and the processes by which it does. Furthermore, research has shown that older adolescents are already beginning to take part in risk behaviors such as drug and alcohol use and premarital sexual behaviors less frequently not as a result of religiosity but due to cultural changes that are leading to less in-person contact (Twenge and Park 2019). It will be interesting to see how religion interacts with these new cultural directions in the lives of older adolescents.

However, though research has clearly shown that religion is associated with several positive adolescent outcomes, this study provides greater clarity regarding which aspects of religiosity are most strongly associated with several outcomes for a sample of young adolescents in Utah. Intrinsic religiosity, though individually related to each outcome ended up having little influence once the other aspects of religiosity and positive youth development were controlled for. One possible reason for this is the nature of the questions which made up the intrinsic spirituality construct. These questions asked about individual spirituality and few subjects indicated high levels even while many indicated higher levels of religious salience and daily religious experiences. Religious salience impacted half of the

positive youth development constructs but seemed to have less direct and indirect impact on the adolescent outcomes themselves.

What appears to make religion most influential in the lives of these adolescents was the feeling that they were actually experiencing divinity in their daily lives (daily religious experiences). Besides the direct effect of daily religious experiences, there was an indirect effect throught the positive youth development concept of connection. Again, this fits well since daily religious experiences indicate that an adolescent felt connected to God. As was pointed out above, this fits well within Brofenbrenner's PPCT model which stressed the importance of proximal processes. Another lens with which to view these findings would be attachment theory. Attachment theory focuses on the relationship and has been applied to religious relationships (relationships with fellow congregants and with God) (Davis et al. 2018). Thus, examining these findings through attachment theory could add further insight on how religion impacts how adolescents navigate challenges. Those who felt that they experience God regularly were less likely to experience anxiety, to be involved in delinquent behaviors. At a time when many adolescents are struggling to avoid the negative consequences so common in society today, it makes sense that there is a need to help them feel greater feelings of connectedness. This appears to hold true for feelings of connectedness with God.

**Author Contributions:** Formal analysis, W.J.D.; Writing—original draft, M.A.G.; Writing—review & editing, M.A.G. and W.J.D. All authors have read and agreed to the published version of the manuscript.

**Funding:** This research received no external funding.

**Institutional Review Board Statement:** This study was approved by the Institutional Review board at Brigham Young University, most recent approval: Review Board Number IRB2021-342.

**Informed Consent Statement:** All subjects signed informed consent statements prior to their participation in the study.

**Data Availability Statement:** For access to the original data, please send a request to the second author (Justin Dyer) at justindyer@byu.edu.

**Conflicts of Interest:** The authors declare no conflict of interest.

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
