# Peer review of "How Does Religious Faith Impact Positive Youth Outcomes"

_religions, doi:10.3390/rel14070881_

Round 1

Reviewer 1 Report

This is an interesting piece which brings additional knowledge about religiosity and the well-being and behaviours of adolescents, e.g. delinquency and anxiety. I think the paper should be published. Still, a few minor issues should be addressed before publishing.

While presenting data on adolescent religiosity, some new data should also be considered, as they suggest a decline of religiosity in the USA and among adolescents.

The literature review clarifies that there are opposite results from various studies about the relationship between religiosity and social outcomes. It is said that this, in many cases, reflects sample and context. This should be elaborated more clearly. i.e. explain how the study's specificities influence results in various cited papers.

This is also important for this study. Limitations are stated in the concluding part of the paper. Still, there is a need to discuss more thoroughly that it is not only about the characteristics of the lack of sample diversity: whether there are single parents or not, whether there are poor people or not, etc.) but of the very fact that the study is mainly about Mormons. How Mormons understand and live religiosity may differ significantly from other religions. However, this brings an opportunity to discuss results in the light of other studies about Mormons, for example, the health outcomes of their religiosity. I do not know if there is similar research about adolescents, but the article would benefit from a general discussion about research on the social impact of religiosity on Mormons. 

Author Response

Response to Reviewer One (Original review in black – author response in blue)

This is an interesting piece which brings additional knowledge about religiosity and the well-being and behaviours of adolescents, e.g. delinquency and anxiety. I think the paper should be published. Still, a few minor issues should be addressed before publishing.

Thank you

While presenting data on adolescent religiosity, some new data should also be considered, as they suggest a decline of religiosity in the USA and among adolescents.

I added a sentence and a source to make this point in the literature review section 1.1

“Despite clear evidence that religiosity has declined among adolescent (Hardy & Longo, 2021), religiosity continues to play a role for the majority of American adolescents.”                                                  

The literature review clarifies that there are opposite results from various studies about the relationship between religiosity and social outcomes. It is said that this, in many cases, reflects sample and context. This should be elaborated more clearly. i.e. explain how the study's specificities influence results in various cited papers.

The current literature review seems to already provide the information requested by the reviewer. The following paragraph provides specific examples of how a study’s specificities (such as which aspect of religiosity) influence results already:

“Though there is a clear correlation between adolescent religiosity and a general re-duction in externalizing behavior, when processes or reasons are explored the findings at times conflict or at minimum show a complicated relationship. For example one study indicated that the protective correlation was stronger when considering how an adoles-cent viewed their relationship with God (relational religiosity) than it was if only looking at adolescents religious activity (Pickering & Vazsonyi, 2010). So an adolescent who felt close to God was less likely to engage in harmful externalizing behavior than one who simply went to church or engaged in other religious activities. However, other studies pointed to the power of those exact activities as the primary reason for the relationship (Moreau, Trussell, & Bajos, 2013). Another study found that the importance of religion to adolescents had little impact on adolescents but church attendance itself did (Mason & Windle, 2002). Clearly there is a need for more research to understand the processes and mechanisms involved. Typically, it has been assumed that religiosity impacts externaliz-ing behavior. But the inverse needs to be considered as well: how does externalizing be-havior impact the development of religiosity. The relationship is likely reciprocal.”

This variation is then hypothetically connected through the theoretical lens of Bronfenbrenner’s PPCT model as explained in these two already existing paragraphs:

“Although as the above literature review indicates, research suggests adolescent relig-iosity significantly influences several outcomes (usually in a positive direction), this in-fluence likely varies person to person as well as circumstance to circumstance. Bron-fenbrenner’s (1999) bioecological model proposes that “proximal processes” (i.e., direct, reciprocal, enduring, increasingly complex interactions) influence both developmental and other outcomes. However, “the form, power, content, and direction of the proximal processes affecting development vary systematically as a joint function of the characteris-tics of the developing person . . . [and] the environment” (Bronfenbrenner, 1999, p. 5). In other words, proximal processes will vary in their effects depending on personal and en-vironmental factors. Thus, the bioecological model would predict differences in the degree to which adolescent religiosity impacts adolescent outcomes dependent upon (a) presence of proximal processes and (b) the characteristics of the child, and their environment.

Proximal processes are the first of four interrelated components of Brofenbrenner’s process-person-context-time (PPCT) model for understanding human development. Ra-ther than believing that individual variables act independently, or even only additively on an adolescent, the PPCT model emphasizes the importance of examining each aspect of development (proximal process, personal characteristics, external context, and time) sim-ultaneously to see how they interact. Accordingly, it is important to understand how sev-eral process-person-context-time constructs may impact adolescent outcomes.

With the literature review already taking up approximately 1/3 of the research article, I hesitate to add much more detail than this.

This is also important for this study. Limitations are stated in the concluding part of the paper. Still, there is a need to discuss more thoroughly that it is not only about the characteristics of the lack of sample diversity: whether there are single parents or not, whether there are poor people or not, etc.) but of the very fact that the study is mainly about Mormons. How Mormons understand and live religiosity may differ significantly from other religions. However, this brings an opportunity to discuss results in the light of other studies about Mormons, for example, the health outcomes of their religiosity. I do not know if there is similar research about adolescents, but the article would benefit from a general discussion about research on the social impact of religiosity on Mormons. 

The following was added to the limitations section where the manuscript discusses the connection to the Mormon church and how that may impact interpretion:

“Mormon adolescents tend to have higher levels of religiosity as measured by church at-tendance, feeling close to God and lower levels of doubt and disaffiliation (See Smith, 2005). This may make the findings most generalizable to other faith traditions such as black and conservative protestant denominations which also tend to have higher levels of religiosity among their adolescent members.”

Reviewer 2 Report

Understanding the mechanisms by which we can mitigate the deleterious effects of stress, anxiety, and delinquency among adolescents constitutes a research program worthy of our sustained attention and investment. This particular paper, as I understand it, proposes that religiosity and the positive youth developmental construct of connection both contribute to such mitigation. The paper further suggests that there are three dimensions to adolescent religiosity, that is, religious salience, intrinsic religiosity, and daily religious experiences. The results of the studies suggest that only daily religious experiences appear to have an impact on internalization and externalization concerns and problems. Daily religious experiences involve a perception of connection to God. You conclude that it would be wise for us to help nurture more feelings of connectedness among adolescents. One possible avenue to accomplish such a goal would apparently be to incentivize beliefs and perceptions regarding connection to a personal god(s). While that may be the case, I’m not sure the costs of incentivizing superstitions, e.g., promoting intellectual dishonesty, is worth it.

I will not comment on the research design itself or its methods. I will say that I did not find Figures 1 & 2 terribly helpful. Also, the paper has some grammatical errors.

Substantively, I found it interesting that the other two dimensions of religiosity examined here did not appear to affect stress, anxiety, and delinquency. While that appears to be the case, I believe you could have described religious salience and intrinsic religiosity a bit more in the introduction in order to further distinguish these from daily religious experience. I believe you correctly identify that your sample ages were not exhaustive of adolescence (FYI, at one point you say the children are between 12 and 14, at another place you say 12 and 15). As I understand the data, generally speaking, there is a tendency for adolescents to feel more stress, anxiety, and isolation a bit later in life; perhaps you could run your experiments again with those between 16 and 18. And, yes, it would be good to get a more varied sample. All of that being said, my bigger concern regards your choice of theory: you positively adopt Brofenbrenner’s PPCT (proximal process-person-context-time) theory. Alas, I am unfamiliar with Brofenbrenner. The bibliography notes three entries for Brofenbrenner, one of which is as an editor. In this regard, your theoretical edifice is predicated on two chapters in edited volumes. I raise this issue because I wonder how received his bioecological model is in the larger field. What is glaringly missing here, at least as far as I can tell, is a consideration of attachment dynamics. The ability of attachment theory to explain variations in adolescent religiosity is widely attested in the literature. To be sure, attachment theory directly studies the dynamics and perceptions of connections between children and parents, and people and god(s). I would argue that attachment is the key, proximal process in coming to understand the dynamics identified in this study. Indeed, if connection is the mediating element, then I believe you would be well advised to consider attachment theory, especially as it addresses the extent to which an adolescent’s perception of God corresponds with or compensates for relationships with caregivers. The work on internal working models (IWMs) would greatly enhance the explanatory value of your results. Although I haven’t read Brofenbrenner, I cannot imagine any bioecological theory that addresses human development not incorporating the robust results of experiments designed to assess attachment theory.

On the whole, the English Language in this article is good. There are a few grammatical mistakes.

Author Response

Response to Reviewer Two

Understanding the mechanisms by which we can mitigate the deleterious effects of stress, anxiety, and delinquency among adolescents constitutes a research program worthy of our sustained attention and investment. This particular paper, as I understand it, proposes that religiosity and the positive youth developmental construct of connection both contribute to such mitigation. The paper further suggests that there are three dimensions to adolescent religiosity, that is, religious salience, intrinsic religiosity, and daily religious experiences. The results of the studies suggest that only daily religious experiences appear to have an impact on internalization and externalization concerns and problems. Daily religious experiences involve a perception of connection to God. You conclude that it would be wise for us to help nurture more feelings of connectedness among adolescents. One possible avenue to accomplish such a goal would apparently be to incentivize beliefs and perceptions regarding connection to a personal god(s). While that may be the case, I’m not sure the costs of incentivizing superstitions, e.g., promoting intellectual dishonesty, is worth it.

I have to admit to being unsure how to interpret the reviewers’ last two sentences in this first paragraph. We clearly agree that no one should encourage superstitions and intellectual dishonesty. However, nothing in this paper examines the truth claims regarding any particular religion or belief in God. This study found that feelings of closeness to others and to God were associated with positive outcomes both by way of reducing externalizing and internalizing problematic behaviors and those finding were reported. Equating encouraging adolescents’ connection to others and God to “incentivizing superstitions and promoting intellectual dishonesty” seems a strange commentary for a journal based on studying religion. I am going to assume this was the reviewer’s perspective and not the editorial direction of the journal since many many articles in the journal explore religion from a social science perspective (which is what this article does) rather than as nothing other than a superstition based in intellectual dishonesty.

I will not comment on the research design itself or its methods. I will say that I did not find Figures 1 & 2 terribly helpful. Also, the paper has some grammatical errors.

Though not requested, we recreated figures 1 and 2 in an attempt to add clarity to research design by removing the variable names and replacing them with clearer descriptive labels and then by combining the two figures into one in an attempt to simplify them (see revised manuscript as the figure wouldn't paste here).

 Substantively, I found it interesting that the other two dimensions of religiosity examined here did not appear to affect stress, anxiety, and delinquency. While that appears to be the case, I believe you could have described religious salience and intrinsic religiosity a bit more in the introduction in order to further distinguish these from daily religious experience.

When we introduce religious salience, we include a parenthetical definition (the importance of religion). Though this seems clear, we expanded it to say (how important religion is in the life of the subject). This construct is so heavily researched we are not sure it requires further definition. However, we provide a detailed definition of intrinsic spirituality and daily religious experience (both well-established scales used regularly in research on religiosity) in our methods section:

“Intrinsic Spirituality. To measure the degree to which children have an internalized sense of spirituality, Hodge’s (2003) six item intrinsic spirituality scale was used. Ques-tions were asked regarding the degree to which children felt spirituality impacts their lives and was important to them. Responses were on a 0 – 10 scale with 0 indicating that spir-ituality was less intrinsic and 10 indicating it was more intrinsic. For example, one item asked “Spirituality is:” with 0 being “not part of my life” and 10 being “the master motive of my life, directing every other aspect of my life.” A measurement model was created for intrinsic spirituality with the factor score exported for use in structural models. The measurement model fit the data well (CFI = .999, RMSEA = .030) with all loadings above .67.”

And

Daily religious Experiences. A subscale of the NIA/Fetzer Religion and Spirituality scale (Idler et al., 2003) was used to examine the degree to which children felt some connection with God/spirituality on a daily basis with response categories being 1 = “never or almost never” to 6 = “many times a day.” Items include “I feel God’s presence” and “I am spiritually touched by the beauty of creation.”  Again, factor scores were exported from a measurement model which fit the data well (CFI = 1.00, RMSEA = 0.00; all standardized loadings above .84).”

I believe you correctly identify that your sample ages were not exhaustive of adolescence (FYI, at one point you say the children are between 12 and 14, at another place you say 12 and 15).

Thank you. That error has been fixed to make it 12-14 consistently.

 As I understand the data, generally speaking, there is a tendency for adolescents to feel more stress, anxiety, and isolation a bit later in life; perhaps you could run your experiments again with those between 16 and 18. And, yes, it would be good to get a more varied sample.

That understanding is correct. Anxiety and stress do tend to increase with age and that will be an excellent future study.

All of that being said, my bigger concern regards your choice of theory: you positively adopt Brofenbrenner’s PPCT (proximal process-person-context-time) theory. Alas, I am unfamiliar with Brofenbrenner. The bibliography notes three entries for Brofenbrenner, one of which is as an editor. In this regard, your theoretical edifice is predicated on two chapters in edited volumes. I raise this issue because I wonder how received his bioecological model is in the larger field. What is glaringly missing here, at least as far as I can tell, is a consideration of attachment dynamics. The ability of attachment theory to explain variations in adolescent religiosity is widely attested in the literature. To be sure, attachment theory directly studies the dynamics and perceptions of connections between children and parents, and people and god(s). I would argue that attachment is the key, proximal process in coming to understand the dynamics identified in this study. Indeed, if connection is the mediating element, then I believe you would be well advised to consider attachment theory, especially as it addresses the extent to which an adolescent’s perception of God corresponds with or compensates for relationships with caregivers. The work on internal working models (IWMs) would greatly enhance the explanatory value of your results. Although I haven’t read Brofenbrenner, I cannot imagine any bioecological theory that addresses human development not incorporating the robust results of experiments designed to assess attachment theory.

The reviewer's point that attachment theory would be an excellent theoretical lens with which to study parent influence on adolescent religiosity is completely valid. However, this study includes not only parental influence but less proximal connections such as with community and society, and internal psychological constructs as well as religiosity constructs. Brofenbrenner’s ecological systems theory allows for just such study. One that looks at several layers of influence, from the individual, to the familial to the community to society at large. It allows for the investigation of proximal processes (such as parent / child) as does attachment theory as well Brofenbrenner’s ecological systems theory is one of the most influential and highly studied theories in several social science fields, especially in the realm of developmental psychology and adolescents. A quick Google Scholar search shows 16,000 references in the last two years alone.  It was recently used in a study published in Religions on “Perspectives on Lifespan Religious and Spiritual Development from Scholars across the Lifespan in 2023. It encompasses the proximal processes examined in attachment theory.

However, to address the reviewers’ concern, a couple sentences were added to the conclusion indicating the possible contribution of looking at these issues through a attachment theory perspective as well.

“Another lense with which to view these findings would be attachment theory. Attach-ment theory focuses on the relationship and has been applied to religious relationships (relationships with fellow congregants and with God) (Davis et al., 2018). Thus, examin-ing these findings through attachment theory could add further insight on how religion impacts how adolescents navigate challenges.”

Reviewer 3 Report

The paper is a clear presentation of how religiosity influence the levels of juvenile delinquency and anxiety experienced by the youth as well as how do the elements of PYD mediate that impact.

The method and vocabulary is properly defined and presented to the reader.

What is missing in the data presentation is wheteher the survey was accepted by the Ethical Board of the endorsing institution. Moreover, the paper should be improved by adding more data presentation in figures or charts especially concerning gender-dependent outcomes and the differences in parents' and children reports on externalizing behaviors (point 6.2).

Another remark is about literature review which is worth enriching with the Jean M. Twenge iGen study (Atria Books, 2017, New York), in which the author admits that adolescents aged 15-19 are less likely to drink alcohol than the Millennials who preceded them, or consider physical fighting as risky and pointless, but at the same time the are loosing their religion (being mostly raised by religiously unafiiliated parents). Thus, the reason for the change in behavior is not religon but cultural shift resulting in the prevalience of online rather than face-to-face contact. Discussion or consideration of this argument may improve the article.

And the last point is the title. I'm not sure if the title is not too general. If 86 percent of the sample were Mormon, shouldn't it be at least added to the title that the paper deals with the influence of the Christian religion on youth?

Some minor English editing examples:

line 32 - importance

l. 94 - protects

116 - related

313- relate

Author Response

Response to Reviewer Three

The paper is a clear presentation of how religiosity influence the levels of juvenile delinquency and anxiety experienced by the youth as well as how do the elements of PYD mediate that impact.

The method and vocabulary is properly defined and presented to the reader.

Thank you.

What is missing in the data presentation is wheteher the survey was accepted by the Ethical Board of the endorsing institution.

Thank you. The following were added to the end of the article:

“Institutional Review Board Statement

Informed Consent Statement

Data Availability Statement

Conflicts of Interest

Moreover, the paper should be improved by adding more data presentation in figures or charts especially concerning gender-dependent outcomes and the differences in parents' and children reports on externalizing behaviors (point 6.2).

Since the gendered differentiation actually washed out (made things non-significant) the findings, I am not sure how a chart/figure would add clarity or meaning. My other concern is that the differences in parent and child report has more to do with the study variable than the findings based on the variable and hence would not seem to strengthen the article. We did recreate the chart showing the statistical model used to guide the study and hope that that added clarity helps.

Another remark is about literature review which is worth enriching with the Jean M. Twenge iGen study (Atria Books, 2017, New York), in which the author admits that adolescents aged 15-19 are less likely to drink alcohol than the Millennials who preceded them, or consider physical fighting as risky and pointless, but at the same time the are loosing their religion (being mostly raised by religiously unafiiliated parents). Thus, the reason for the change in behavior is not religon but cultural shift resulting in the prevalience of online rather than face-to-face contact. Discussion or consideration of this argument may improve the article.

We love Twenge’s book but as the reviewer points out – it is based in older adolescents (15-19) and this study is based in younger adolescents (12-14). However, I added the following to the conclusion / limitations section to highlight the point the reviewer is making:

“Furthermore, research has shown that older adolescents are already beginning to take part in risk behaviors such as drug and alcohol use and premarital sexual behaviors less fre-quently, not because of religiosity but, because of cultural changes that are leading to less in person contact. (Twenge & Park, 2019). It will be interesting to see how religion interacts with these new cultural directions in the lives of older adolescents.”

And the last point is the title. I'm not sure if the title is not too general. If 86 percent of the sample were Mormon, shouldn't it be at least added to the title that the paper deals with the influence of the Christian religion on youth?

I have added this to the title which now reads: “How Does Christian Religious Faith Impact Positive Youth Development” I have to admit that I prefer the original title since most of the constructs involved in this study are also part of many non-Christian religious traditions – especially the Abrahamic religions of Christianity, Islam and Judaism. I will leave it to the editors to decide however, I would keep the original title myself.

Comments on the Quality of English Language

Some minor English editing examples:

line 32 – importance

Fixed

  1. 94 – protects

Fixed

116 - related

Fixed

313- relate

Fixed